# Targeting ARID1A-Deficient Cancers: An Immune-Metabolic Perspective

**DOI:** 10.3390/cells12060952

**Published:** 2023-03-21

**Authors:** Timofey Lebedev, Rubina Kousar, Bbumba Patrick, Muhammad Usama, Meng-Kuei Lee, Ming Tan, Xing-Guo Li

**Affiliations:** 1Department of Cancer Cell Biology, Engelhardt Institute of Molecular Biology, Russian Academy of Sciences, 119991 Moscow, Russia; lebedevtd@gmail.com; 2Graduate Institute of Biomedical Sciences, China Medical University, Taichung 110122, Taiwan; rubimansib@gmail.com (R.K.); bumbax22@gmail.com (B.P.); usamakhan850@gmail.com (M.U.); laviyu0111@gmail.com (M.-K.L.); mingtan@cmu.edu.tw (M.T.); 3Research Center for Cancer Biology, China Medical University, Taichung 110122, Taiwan; 4Institute of Biochemistry and Molecular Biology, College of Life Sciences, China Medical University, Taichung 110122, Taiwan

**Keywords:** ARID1A, metabolism, cancer immunotherapy, chromatin remodeling

## Abstract

Epigenetic remodeling and metabolic reprogramming, two well-known cancer hallmarks, are highly intertwined. In addition to their abilities to confer cancer cell growth advantage, these alterations play a critical role in dynamically shaping the tumor microenvironment and antitumor immunity. Recent studies point toward the interplay between epigenetic regulation and metabolic rewiring as a potentially targetable Achilles’ heel in cancer. In this review, we explore the key metabolic mechanisms that underpin the immunomodulatory role of AT-rich interaction domain 1A (ARID1A), the most frequently mutated epigenetic regulator across human cancers. We will summarize the recent advances in targeting ARID1A-deficient cancers by harnessing immune-metabolic vulnerability elicited by ARID1A deficiency to stimulate antitumor immune response, and ultimately, to improve patient outcome.

## 1. Introduction

Epigenetic alterations and metabolic rewiring are among the well-known cancer hallmarks. Increasing evidence demonstrates that epigenetic remodeling and metabolic reprogramming are highly intertwined and reciprocally regulate each other, and their perturbations often enhance cancer progression and metastasis, as well as reshape the tumor microenvironment (TME) to drive immune escape [1]. On the one hand, oncogenic signaling-driven metabolic rewiring could alter the epigenetic landscape of cancer cells. On the other hand, epigenetic remodeling could influence the expression of metabolic genes. There are recent excellent reviews on the role of the epigenomic-metabolic interplay in cancer cells [2,3,4]. Intriguingly, emerging insights into the hostile TME reveal that antitumor immune response can be dynamically regulated through epigenetic and metabolic-based reprograming, which may be exploited as a potentially targetable Achilles’ heel in cancer [5,6,7,8,9,10]. Therefore, a further understanding of the links between epigenetic modifications and cellular metabolism has the potential to uncover novel therapeutic interventions for enhanced antitumor immunity. 

AT-rich interaction domain 1A (ARID1A) is the DNA-binding subunit of the SWI/SNF (switch/sucrose non-fermentable) chromatin remodeling complex, which regulates chromatin accessibility by using energy provided by ATP hydrolysis and modulates a wide range of cellular processes, including transcription, replication and DNA damage repair [11]. Functional genomics studies have revealed that genes encoding subunits of SWI/SNF are collectively mutated in 20–25% of all human malignancies, among which ARID1A is the most frequently mutated subunit in a broad spectrum of human cancers [11] (Table 1). Recent mechanistic studies have further demonstrated a crucial role for ARID1A in the regulation of gene expression that drives oncogenesis or tumor suppression [12,13,14]. The mechanism of action of ARID1A in cancer has been mainly attributed to its chromatin remodeling function within the SWI/SNF complex, leading to aberrant cell proliferation, differentiation and apoptosis that have tumorigenic consequences [11]. Intriguingly, emerging studies highlight a critical role for ARID1A in controlling metabolic circuits combined with epigenetic remodeling to shape the functional states of immune cells (Figure 1). Here, we summarize recent progress on the importance of ARID1A in orchestrating epigenetic remodeling and cellular metabolism to influence antitumor immunity. We will focus in particular on emerging strategies for the targeted therapy of ARID1A-deficient cancers by exploiting immune-metabolic vulnerability elicited by ARID1A loss to enhance antitumor immune response.

## 2. ARID1A: Context-Dependent Functions

Arid1a is widely expressed across a large majority of cell types during multiple stages of mouse embryonic development [24]. Ablation of murine *Arid1a* in early mouse embryos causes the absence of the mesodermal layer, developmental arrest and embryonic lethality (at about embryonic day 6.5). Specifically, loss of *Arid1a* impairs the pluripotency and self-renewal of embryonic stem (ES) cells, leading to cell lineage-specific differentiation defects, where *Arid1a*-ablated ES cells are defective in differentiating into mesoderm-derived cardiomyocytes and adipocytes but not ectoderm-derived neurons. Further mechanistic analysis indicates that ARID1A regulates the expression of genes essential for ES cell self-renewal, differentiation and cell linage decisions through chromatin remodeling [24]. These results suggest that ARID1A may have cell lineage-dependent functions. 

Consistently, tissue-specific knockout mouse models of murine *Arid1a* have been reported in different compartments, displaying context-dependent phenotypes. In microglia, the loss of *Arid1a* leads to the perturbation of microglial homeostasis through disruption of genome-wide H3K9me3 occupancy and changes of the chromatin landscape of regulatory elements that influence the switching of microglial states [25]. Furthermore, microglia-specific *Arid1a* ablation enhances M1 microglial polarization and weakens M2 polarization, which may explain anxiety-like behaviors in these animals [26]. In the liver, hepatocyte-specific abolition of *Arid1a* impairs liver regeneration through down-regulation of chromatin accessibility and gene expression associated with injury-induced liver-progenitor-like cells [27]. Furthermore, in the thymus, conditional deletion of *Arid1a* in T-cell lineage leads to developmental blocks from double-negative to double-positive stages mainly through reducing intracellular TCRβ expression and inducing apoptotic pathways [28]. Finally, hematopoietic cell-specific deletion of *Arid1a* impairs the differentiation of both myeloid and lymphoid lineages through a global reduction in open chromatin and gene transcription of key genes involved in hematopoietic development [29]. 

Given the crucial role of ARID1A in multiple tissue types, it is not surprising to see that mutations of ARID1A have emerged in a broad range of cancers of different origins. According to a recent analysis that integrates epidemiological and genomic data to provide real-world estimates for the overall proportion of US cancer patients who harbor a coding exonic mutation in each gene, ARID1A is among the ten most commonly mutated driver genes, highlighting the role of epigenetic dysregulation in cancer [30]. Furthermore, whole exome sequencing (WES) of biopsies from 292 patients with metastatic solid tumors enrolled in the Gustave Roussy-sponsored MOSCATO (NCT01566019) and MATCHR (NCT02517892) trials shows that ARID1A is among the most frequently altered genes in metastatic solid tumors [31]. Intriguingly, recent studies indicate age-dependent mutation patterns for ARID1A. In Burkitt lymphoma, in contrast to TP53 and CCND3, mutation frequencies of which are not age related, samples from pediatric patients showed a higher frequency of mutations in ARID1A than those from adults [32]. Similarly, comparison of younger to older women with luminal A tumors revealed significant differences in somatic alterations in three prevalent driver genes. PIK3CA alterations were more common in older patients, whereas GATA3 and ARID1A alterations were more common in younger patients [33]. Together, the different frequencies of ARID1A mutations in different age groups may indicate different biological susceptibilities to ARID1A alterations, adding additional layer of complexity to the cell type, and age-dependent functions of ARID1A.

To differentiate the context dependence on ARID1A in cancers carrying either wildtype or mutant ARID1A, we analyzed the cell type dependencies on ARID1A across 32 cancer types. As shown in Figure 2, cancer cells with ARID1A mutations tend to be less affected by ARID1A depletion than those without mutations. This result is in agreement with the notion that most mutations of ARID1A reported so far tend to lead to its loss of function, thus making ARID1A to be considered as a tumor suppressor [34]. Importantly, in endometrium-related uterine and ovarian carcinomas exhibiting the highest mutation efficiency of ARID1A, it appears that dependence on ARID1A is also observed in cancers carrying ARID1A mutations, suggesting that ARID1A may exert pro-tumor functions in certain subtypes of these cancers, where mutation of ARID1A may have antitumor activity. This context-dependent role of ARID1A is also seen in other cancers, such as lymphoma, colon cancer, bladder cancer, pancreatic cancer and liver cancer (Figure 2).

Consistent with the pro-tumor functions of ARID1A under certain contexts, such as hepatocellular carcinoma (HCC), the tumors have higher expression levels of ARID1A than adjacent normal tissues [39]. Moreover, in mice with liver-specific *Arid1a* manipulation, loss of *Arid1a* presents resistance to tumor initiation whereas ARID1A overexpression accelerates initiation. However, in established liver tumors, *Arid1a* loss promotes tumor progression and metastasis. These results provide clear in vivo evidence supporting context-dependent tumor suppressive and oncogenic role of ARID1A [40]. Consistently, endometrial cancers have wild-type ARID1A in the primary tumors, whereas the metastatic subclones carry deleterious mutations, further indicating context-dependent function of ARID1A in primary versus metastatic cancers [41].

## 3. ARID1A: Synthetic Lethality and Beyond

Synthetic lethality was initially described in Drosophila as recessive lethality [42]. Classically, it is defined as a setting where loss of either of two genes independently has litter effect on cell viability, whereas loss of both genes simultaneously causes cell death. In cancer, the idea of synthetic lethality has been extended to gene pairs, where pharmacological inhibition of gene A leads to death of cancer cells carrying loss of gene B through mutation or deletion, whereas normal cells carrying an intact gene B lack the genetic interaction between gene A and gene B and therefore are spared the effect of the drug [43]. It is important to note that many synthetic lethal pairs are likely to be context dependent. This could be due to the specific contexts where alterations of additional genes or pathways may alter the functional interactions of the synthetic lethal pair. In addition, as the landscape of functional genetic lesions, such as driver mutations, vary among different tissues of origin, or among different subgroups of the same cancer, these genetic contexts contribute to the selective dependency of synthetic lethal gene pairs [43]. Recent advances, including CRISPR/Cas9-based gene editing, have made possible systematic screens for synthetic lethal targets in human cancers harboring mutations of tumor suppressor, such as ARID1A. Ever since the discovery as a tumor suppressor for ARID1A, synthetic lethal phenotypes have been uncovered in ARID1A-deficient cells when treated with PARP inhibitors, ATR inhibitors, EZH2 inhibitors, and other emerging agents [13].

ARID1B, a paralogue of ARID1A, is among the top genes first identified that are preferentially required for the survival of ARID1A-mutant cancer cells [44]. Molecularly, ARID1B is mutually exclusive with ARID1A in the SWI/SNF complex. Recent studies further point out the developmental switch from ARID1A- to ARID1B-containing SWI/SNF complexes in neural crest development [45]. In ARID1A-deficient cancers, at least one functional ARID1B allele is maintained, and loss of ARID1B leads to destabilization of the SWI/SNF complex and subsequent impaired cell proliferation [44]. To illustrate the context-dependency of synthetic lethality, we further analyzed the cell type dependencies on ARID1B in cancers with or without ARID1A-deficinecy across 32 cancer types. Similar to the scenario of cell-type dependence on ARID1A, cancers displaying the highest ARID1B dependency score (lower than −0.5 or so) tend to have ARID1A mutations, further validating the close functional interaction between these two core subunits of the SWI/SNF complex (Figure 3). Not surprisingly, this synthetic lethality relationship appears to be absent in certain subtypes of cancers harboring ARID1A mutations, such as endometrial/uterine cancer, pancreatic cancer, ovarian cancer, lung cancer and colon cancer (Figure 3). Furthermore, in cancer cell lines from the CCLE database, the frequency of ARID1A damaging mutations ranged from 0%, for example for head and neck cancer cells, up to 62% (23 out of 37 cell lines have ARID1A mutations) for endometrial and uterine cancer cells (Appendix A). We also found high frequency of ARID1A mutations among bladder (8/31), ovarian (14/61), pancreatic (10/48), lymphoma (8/41), colorectal (12/67) and gastric (7/40) cancer cell lines. Among cell lines with ARID1A mutations, 1.6% (2/123) were dependent on ARID1A, with dependency scores < −0.5, and among cell lines without ARID1A mutations, 3.8% (39/1035) were dependent on ARID1A expression (Figure 2, Appendix A). Notably, 16.8% (18/107) of cell lines with ARID1A mutations, but only 0.19% (2/1072) of cell lines without ARID1A mutations were dependent on ARID1B expression (Figure 3, Appendix A). These results suggest that future studies are required to identify more genetic contexts that allow the discovery of cancer type/subtype-specific synthetic lethal interactions and novel cancer drug targets. 

## 4. ARID1A-Deficiency: An Immune-Metabolic Vulnerability 

Recent advancements in cancer immunology have shifted the paradigm of cancer treatment and has become one of the most prominent therapeutics for human cancers. This breakthrough has prolonged the survival of patients with relapsed or refractory metastatic cancers; however, only a subset of patients display favorable responses and most patients with immunologically cold solid tumors do not respond [46,47]. Emerging evidence has demonstrated that productive antitumor immune response is regulated in multiple cell types of TME and at an extraordinary level of orchestration of an array of coordinated biological processes, such as epigenetic modifications and metabolic reprograming. The role of the metabolic-epigenetic axis in antitumor immunity has been extensively reviewed elsewhere [48,49]. Given the prominent role of ARID1A in cancer, iterative insights into how to target ARID1A-deficient cancers by harnessing the immune-metabolic vulnerability are critically needed. Here, we summarize the current understanding of the epigenetic and metabolic mechanisms underlying the antitumor immune response in ARID1A-deficient cancers. This could have important implications for the identification of novel drug targets and the development and application of rational combination therapies. 

### 4.1. Metabolic Rewiring in ARID1A-Deficient Cancers

Ogiwara et al. reported the first study linking metabolic dependency and ARID1A deficiency in cancer [50]. In this study, a drug-sensitivity screen in ARID1A-deficient cancer cells identified unique hits: PRIMA-1 and APR-246; both of which inhibit the activity of the antioxidant metabolite glutathione (GSH) by covalent binding to thiols. Importantly, this selective sensitivity was observed in multiple ARID1A-deficient cancers, irrespective of the cellular context. Further mechanistic investigations demonstrate that this vulnerability in cancer cells lacking ARID1D results from low basal levels of GSH due to reduced expression of SLC7A11 caused by impaired ARID1A-mediated chromatin remodeling and transcriptional activation. Since SLC7A11 encodes a subunit of the cystine/glutamate transporter XCT that supplies cells with cysteine, a key source of GSH, ARID1A-deficient cancer cells are specifically vulnerable to inhibitors of the GSH metabolic pathway, such as APR-246 for GSH and buthionine sulfoximine for glutamate-cysteine ligase catalytic subunit (GCLC), an essential subunit for ATP-dependent enzyme glutamate-cysteine ligase synthetase (GCL) that catalyzes the rate-limiting step of glutamate ligation with cysteine during GSH synthesis [50]. Given that antioxidant defense systems have been considered promising targets in cancer therapy, depletion of cysteine was shown earlier as an alternative antioxidant pathway to confer metabolic dependency on thioredoxin in breast cancer cells [51]. Thus, the link between ARID1A and synthesis of reduced GSH provides a targetable metabolic vulnerability of ARID1A-inactivating mutations, a genetic aberration commonly observed in a variety of human malignancies. 

The notion of metabolic dependency elicited by ARID1A deficiency was further affirmed by recent studies. In ARID1A-inactivated ovarian carcinoma clear cell carcinoma (OCCC), the loss of ARID1A increases glutamine utilization and metabolism through upregulation of the expression of glutaminase (GLS1), a key enzyme in glutamine metabolism that is overexpressed in several human cancers. Therefore, this dependence on glutamine metabolism created by ARID1A inactivation could be therapeutically exploited by pharmacological inhibition of glutaminase with an inhibitor CB-839, which selectively suppresses the growth of ARID1A mutant cells. Furthermore, CB-839 synergizes with immune checkpoint blockade (ICB) anti-PDL1 antibody in a genetic mouse model driven by conditional Arid1a ablation, suggesting a novel combination therapy to target ARID1A-deificent OCCC [52]. Given a forthcoming clinical trial testing CB-839 in combination with a PARP inhibitor in platinum resistant ovarian cancer (NCT03944902) [53], a more recent study addressed the question of whether GLS1 could serve as a predictive biomarker for the therapeutic response of OCCC to CB839. Clemente et al. found no correlation between GLS1 overexpression and ARID1A status in clinical specimens of OCCC, raising the caution that should be taken when considering the use of CB839 to treat OCCC patients with ARID1A mutations [54]. Taken together, on the one hand, these findings may seem controversial. On the other hand, these results may point toward the context-dependent role of ARID1A regarding the control of GLS1 expression and sensitivity of ARID1A-deficient cells to GLS1 inhibition. 

In line with context-dependent functions of ARID1A, a recent study in HCC provides additional insights into the link between epigenetic regulatory function of ARID1A to cancer metabolism, which may offer a new possibility of targeting the metabolic vulnerability in certain HCC patients. Somatic mutations of ARID1A occurs in around 10% of patients with HCC [55]. Zhang et al. found that knockout of ARID1A in HCC cells did not promote cell proliferation under the normal culture condition but empowered growth advantage in a glucose-deprived condition. Further analysis demonstrated that ARID1A loss provides protection from glucose deprivation-induced cell death. Mechanistically, AIRD1A recruits another epigenetic regulator, HDAC1 to the promoter of USP9X, resulting in down-regulation of USP9X and its target protein kinase AMP-activated catalytic subunit α2 (AMPKα2), an isoform of AMPK. As a consequence, upregulation of USP9X due to ARID1A inactivation reduces the ubiquitination-associated protein degradation, stabilizes AMPKα2 and thus activates AMPK signaling, a key energy sensing pathway that reprograms multiple metabolic pathways, such as glycolysis, lipid metabolism, and antioxidant production, to provide a growth advantage to HCC cells under the glucose-deprived condition. Importantly, ARID1A-inactivated cells are more sensitive to treatment with Compound C, an AMPK inhibitor, as shown by a prolonged survival of tumor bearing mice [56]. It is important to note that as revealed by the RNA-seq data of the TCGA HCC cohort, ARID1A mRNA is significantly upregulated in human HCC tumors as compared with the nontumorous liver samples [39], in contrast to the frequent downregulation of ARID1A protein in other ARID1A-deficient cancers. These results imply that there exist multiple levels of regulation of the expression of ARID1A, most likely at the post-transcriptional or post-translational levels. Future studies are warranted to investigate how dysregulation of ARID1A protein translation and turnover may contribute to ARID1A-related oncogenesis. Furthermore, given the multiple faces of AMPK signaling in cancer, AMPK activator metformin exhibits anticancer effects in various cancers, including HCC [57]. Although these findings may seem controversial, rather, they support the idea that the heterogeneous context of a tumor may account for the paradoxical role of either oncogenic or tumor suppressive functions of the same protein.

Finally, recent studies have implicated mitochondrial and metabolic reprogramming as new vulnerabilities of ARID1A deficiency. Srinivas et al. demonstrated that cells lacking ARID1A are highly sensitive to inhibition of polo-like kinase 1 (PLK1) [58]. PLK1, a serine-threonine kinase, plays a critical role in cell cycle progression, cell division and DNA damage response and overexpression of PLK1 promotes cell proliferation [59]. However, despite the observations that PLK1 inhibition selectively reduces cell proliferation in cancer cells but not in normal cells, the clinical efficacy of PLK1 inhibitor is limited due to a lack of biomarkers that could be used to stratify patients who will respond to PLK1 inhibitor. In Srinivas et al.’s study, the loss of ARID1A altered mitochondrial biogenesis characterized by a higher number of globular mitochondria, increased oxidative phosphorylation, and elevated oxygen consumption without ATP production. Intriguingly, given the mitochondrial localization of PLK1, PLK1 inhibition leads to enhanced oxygen consumption, and thus membrane depolarization of mitochondria and eventually cell apoptosis. Taken together, ARID1A inactivation creates a unique vulnerability of mitochondrial metabolism involving PLK1, the inhibitor of which has been approved by FDA in acute myeloid leukemia [59]. Consistent with the objective response of PLK1 inhibition in a subset of refractory cancers, a combination of PLK1 inhibitor and temozolomide, an alkylating agent prodrug, shows synergistic cytotoxicity in glioma cells [60]. Considering the growing list of synthetic lethal gene pairs with ARID1A deficiency, it may be interesting to investigate the efficacy of combinational therapy of PLK1 inhibitor and agents targeting synthetic lethal phenotypes in ARID1A-mutated cancers, such as PARP inhibitors and ATR inhibitors. Furthermore, by using a genetically engineered mouse model of lung adenocarcinoma by ablating Smarca4 in the lung epithelium, Deribe et al. demonstrated that Smarca4, as a bona fide tumor suppressor, cooperates with p53 loss and Kras activation through enhanced oxygen consumption and increased respiratory capacity. Interestingly, SMARCA4 mutant lung cancer cells and xenograft tumors have marked sensitivity to inhibitors of oxidative phosphorylation (OXPHOS), reagents currently under clinical development [19]. Consistent with the idea of harnessing metabolic vulnerability in ARID1A-deficient lung cancers, a recent study reveals that ARID1A loss promotes lung tumorigenesis through enhanced glycolysis. Importantly, treatment with the small bromodomain and extraterminal protein (BET) inhibitor JQ1 compromised the initiation and progression of ARID1A-deficient lung adenocarcinoma [61]. Together, these results suggest that metabolic reprogramming caused by ARID1A mutations could confer therapeutically exploitable synthetic lethal interactions. 

### 4.2. ARID1A as a Modulator of Antitumor Immunity

Despite the overall impressive clinical effects of ICB in cancer treatment, not all patients respond to immunotherapy. Therefore, there is a substantial and unmet medical need for the development of biomarkers that could predict the response of individual patients to immunotherapeutic agents [46,47]. Recent findings that ARID1A deficiency is related to sensitivity to ICB therapy in multiple cancer types have paved the way of harnessing ARID1A as a novel target for cancer immunotherapy [11,62,63,64,65,66,67,68,69,70,71,72]. Furthermore, these results suggest that the status of ARID1A may serve as a companion biomarker for efficacy prediction [11]. Importantly, more recent studies have provided new insights into the role for AIRD1A in modulating antitumor immunity, allowing the development of mechanism-based rational combination therapy to potentiate the efficacy of ICB in ARID1A-deficent tumors [73,74]. 

Shen et al. reported a positive correlation between ARID1A mutations and enhanced antitumor immunity in ovarian tumors [62]. ARID1A interacts with mismatch repair (MMR) protein MSH2 and promotes NMR through recruitment of MSH2 to chromatin during DNA replication. Conversely, ARID1A deficiency leads to compromised NMR, which may provide one explanation of the tumor-suppressive role of ARID1A. Intriguingly, ARID1A-deficient tumors display increased mutation load, upregulation of PD-L1 expression, and elevated numbers of tumor-infiltrating lymphocytes (TILs). Notably, ARID1A-mutant tumors are sensitized to treatment with anti-PD-L1 antibody [62]. Consistently, a recent study of 3403 patients receiving ICB therapy across nine distinct cancer subtypes revealed ARID1A alterations as a biomarker for longer progression-free survival (PFS) after immunotherapy [63]. Interestingly, the PFS after ICB therapy was significantly longer in patients with altered ARID1A than in those with wildtype ARID1A, regardless of microsatellite instability (MSI) and tumor mutation burden (TMB) statuses [63]. In line with the immunogenicity of ARID1A mutation, a more recent study in 417 colorectal cancer patients demonstrated that ARID1A mutation may define an immunologically active subgroup in patients with microsatellite stable (MSS) tumors [64]. ARID1A mutation was enriched in immune subtype, where there was a strong correlation between ARID1A status and expression for IFNγ, key checkpoint genes (e.g., PD-L1, CTLA4, and PDCD1) and certain gene sets (e.g., antigen presentation, cytotoxic T-cell function, and immune checkpoints) [64]. The correlation between ARID1A status and PD-L1 expression was further confirmed in a relatively large series of samples from 72 advanced gastric cancer patients [65]. In this study, the expression of PD-L1 in cancer cells was not associated with progression-free survival (PFS) and overall survival (OS), regardless of whether PD-L1 was measured in tumor cells or in the immune infiltrate. Interestingly, in the subgroup analysis, positive PD-L1 immunostaining in tumor cells was associated with better PFS in patients not carrying activation of DNA damage repair components, as defined by the presence of ARID1A mutations and negative pATM expression. However, this correlation was lost in patients whose tumors carry wildtype ARID1A and concomitant pATM expression, thus denoting activation of DNA damage repair pathway [65].

Furthermore, in pancreatic cancer, where ICB has shown little efficacy, a more recent study showed that a small subset of patients with ARID1A alterations achieved an objective response, including a complete remission after immunotherapy treatment, thus warranting prospective trials (ClinicalTrials.gov, NCT02478931) [66]. In contrast, Li et al. demonstrated that ARID1A aberrations led to impaired chromatin accessibility to IFN-responsive genes, limited IFN gene expression, poor tumor immunity and attenuated host survival in both experimental models and patient samples [67]. Mechanistically, ARID1A interacts with EZH2 to antagonize EZH2-mediated IFN responsiveness and immune evasion [67]. Taken together, future efforts in retrospective studies should focus towards identifying novel biomarker combinations to stratify subgroups of patients with ARID1A alterations for better outcomes upon immunotherapy treatment. 

A number of studies have investigated the combination of immunotherapy with agents targeting the synthetic lethal interactions with ARID1A [68,69,70]. Fukumoto et al. demonstrated that inhibition of HDAC6 synergizes with anti-PD-L1 ICB in ARID1A-inactivated ovarian cancer [68]. The same group showed earlier that HDAC6 activity is essential in ARID1A-mutated ovarian cancers [69]. The dependence on HDAC6 activity in ARID1A-mutated cells is attributed to a direct transcriptional repression of HDAC6 by ARID1A whereas ARID1A deficiency abolishes the apoptosis-promoting function of p53 by upregulating HDAC6. Therefore, HDAC6 inhibition selectively promoted apoptosis of cells with ARID1A mutation, but not ARID1A wildtype, as verified by the improved survival of ARID1A-deficient tumor-bearing mice upon treatment of a clinically applicable small molecule inhibitor of HDAC6 [69]. In line with the role of HDAC6 in modulating tumor immune microenvironment, combinational treatment with HDAC6 inhibitor and anti-PD-L1 antibody resulted in reduced tumor burden and improved survival. Molecularly, ARID1A directly repressed the transcription of PD-L1. Thus, the synergy between HDAC6 inhibition and ICB is due to activation and infiltration of IFNγ-positive CD8 T cells [68]. More recently, Goswami et al. revealed ARID1A mutation in tumor cells and expression of immune cytokine CXCL13 in the baseline tumor tissues as two predictors of clinical response to ICB in metastatic urothelial carcinoma [70]. Further, reverse translational studies demonstrated that mice bearing CXCL13-ablated tumors were resistant to ICB, whereas loss of ARID1A improved sensitivity to anti-PD-1 therapy in a murine model of bladder cancer. Importantly, combination of ARID1A and CXCL13 in baseline tumor tissues suggested improved overall survival compared to either single biomarker [70]. These findings suggest that further identification of other relevant biomarkers, together with ARID1A status, will be warranted to enhance the prediction power of patient response to immunotherapy. 

Given the emerging roles of ARID1A in the DNA damage response [13], accumulating evidence has enhanced our understanding of the biological role of ARID1A, offering new mechanisms for synthetic lethality-based targeting of ARID1A-inactivated cancers, such as inhibition of PARP [71,72] and ATR [73]. Recently, Wang et al. reported that ARID1A-mutated/deficient tumors exhibited high expression of Chk2, a DNA damage checkpoint kinase downstream of ATM, through modulating autoubiquitination of the E3-ligase RNF8 and thereby reducing RNF8-mediated Chk2 degradation. Interestingly, inhibition of the ATM/Chk2 DNA damage checkpoint axis resulted in replication stress and accumulation of cytosolic DNA, which subsequently activated the DNA sensor STING-mediated innate immune response in ARID1A-deficient tumors. Importantly, an ATM inhibitor selectively sensitized ICB efficacy in tumors with ARID1A-depletion, but not ARID1A WT [74]. Overall, these findings open up a new avenue to target ARID1A-mutated cancers through selectively modulating cancer cell intrinsic innate immunity to potentiate the antitumor efficacy of immunotherapy.

Finally, until recently, the direct role of ARID1A in CD8+ T cell exhaustion remains elusive. Through an in vitro CRISPR-Cas9 screen, Belk et al. showed that elimination of ARID1A confers CD8+ T cells proliferative and cytotoxic function in vivo, leading to better anti-tumor activity [75]. Depletion of ARID1A resulted in downregulation of exhaustion-related genes in tumor-infiltrating T cells through limiting the acquisition of exhaustion-associated chromatin accessibility [75]. These findings provide additional evidence supporting the continuing idea that epigenetic modification is a key mechanism controlling antitumor immunity in general, and CD8+ T cell exhaustion in particular. However, given the recurring themes that loss of ARID1A in epithelial or hematopoietic cells can be a step toward cancer transformation, these new findings do raise an interesting question of whether targeting ARID1A may increase the transformative potential of T cell to acquire additional cancer characteristics. Importantly, given recent advances in determining T cell subpopulations as predictive biomarkers for immunotherapy through the use of single-cell RNA-sequencing technology [76], future studies may be warranted to carefully evaluate the effects of targeting ARID1A on the antitumor T cell response versus other cancer characteristics of the TME at more improved resolution. Besides T cells, polymorphonuclear myeloid-derived suppressor cells (PMN-MDSCs) have been recently identified as the major immune cell type that leads to an immunosuppressive microenvironment in ARID1A-deficient prostate cancer [77]. Mechanistically, loss of ARID1A, along with PTEN deficiency, promotes prostate tumorigenesis through activation of NF-kB signaling, leading to CXCR2-mediated chemotaxis of MDSCs to shape the immunosuppressive TME, thus suggesting a potential therapeutic strategy of targeting ARID1A/NF-kB/CXCR2 axis in combination with ICB for advanced prostate cancer [77]. 

## 5. Future Outlooks

ARID1A has emerged as a tumor suppressor in a broad array of human malignancies. Over the years, mutation of ARID1A has been extensively explored for synthetic lethal targeting several fundamental cellular functions including DNA damage response and cellular metabolism. The recent discovery of the SWI/SNF complex as an essential player in determining the therapeutic efficacy of cancer immunotherapy further highlights great promise for harnessing the cancer-specific vulnerability elicited by ARID1A deficiency to potentiate immunotherapy. Despite the promise, several challenges remain to be tackled before we realize the full potential of ARID1A-based precision oncology. 

Accumulating evidence has demonstrated promising preclinical and clinical results of combinational therapy to target cancers harboring ARID1A mutations. We summarize the recent clinical trials of targeting ARID1A-deficient cancers in Table 2. To move forward, there remains an urgent need to develop predictive biomarkers that can identify the population of patients most likely to respond to the therapy. However, the seemingly controversial results have been reported regarding the capacity of ARID1A mutations in predicting the efficacy of ICB therapy. Thus, future studies utilizing larger cohorts of patients and longer follow-up durations would be beneficial to determine the clinical association. On the other hand, it is possible that the predictive capacity of ARID1A deficiency for ICB efficacy may be limited to the specific functional context, where other unknown factors contribute to the complexity. In this regard, future studies are warranted to develop a set of biomarkers that can accurately define the specific genetic, metabolic or epigenetic contexts where a single biomarker may lack enough prediction power. In particular, biomarkers that can be readily tested in patient’s body fluids and biopsy specimens will be essential for the appropriate stratification of patients.

In addition to cancer cells, multiple cell types in the TME, including immune cells may be differentially influenced by the metabolic rewiring and epigenetic alterations caused by the driver mutation, such as ARID1A deficiency [78]. Given that most of the current synthetic lethal interactions have been based on the functional associations and phenotypic observations, more in-depth mechanistic understanding will continue to provide key insights into how the differential susceptibility of cancer cells and various immune cells to ARID1A deficiency promotes or suppresses antitumor immunity. To that end, it will be important to determine the functional difference between the SWI/SNF complex in the presence and absence of ARID1A in the context of TME. In the near future, these investigations will help to understand how alteration of the chromatin remodeling activity of SWI/SNF complex contributes to pathogenesis in tumors harboring ARID1A mutations and help discover more effective therapeutic targets. 

## Figures and Tables

**Figure 1 cells-12-00952-f001:**
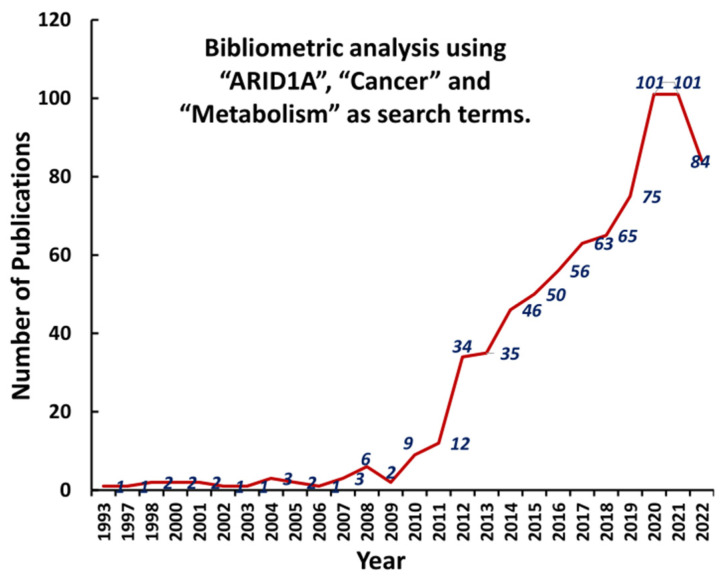
Bibliometric analysis of publications using “ARID1A”, “Cancer” and “Metabolism” as search terms. We searched the PubMed to perform bibliometric analysis, using “ARID1A”, “Cancer” and “Metabolism” as search terms. The number of published articles has increased each year, in particular, during the past 10 years.

**Figure 2 cells-12-00952-f002:**
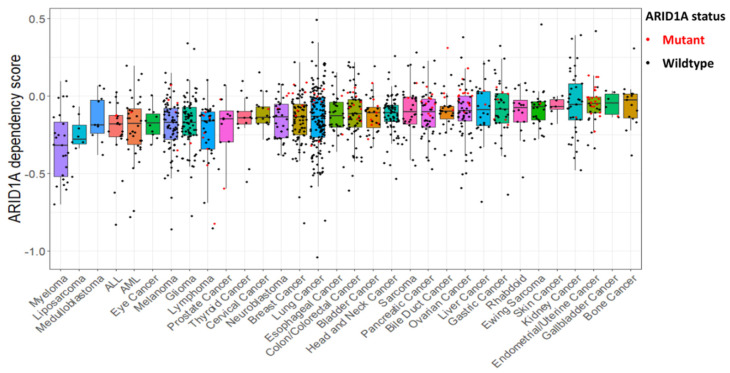
Cancer type-specific ARID1A dependency. Shown is the ARID1A dependency score across 32 cancer types comparing cancers with ARID1A mutations (red dots) versus wildtype (black dots). Dependency scores were calculated for each cell line using data from DepMap database version 22Q2 as an average gene score from RNAi (Achilles + DRIVE + Marcotte, DEMETER2) [35,36] and CRISPR (DepMap 22Q2 Public + Score, Chronos) [37] screens. Negative scores represent more dependent cells, which have reduced proliferation after gene depletion by CRISPR/Cas9 or RNAi. Detailed description is provided in [38]. Data for ARID1A mutations in cancer cell lines was obtained from the Cancer Cell Line Encyclopedia (CCLE) database and only damaging mutations are shown. Note that cancer cells with ARID1A mutations are less affected by ARID1A depletion than those without ARID1A mutations (wildtype ARID1A), as shown by most red dots placed higher than the median values for that cell type.

**Figure 3 cells-12-00952-f003:**
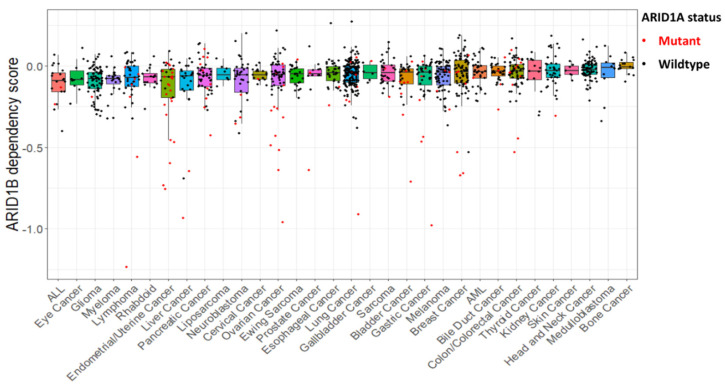
Distribution of cancer type-specific ARID1B dependency. Shown is the ARID1B dependency across 32 cancer types comparing cancers with ARID1A mutations (red dots) versus wildtype (black dots). Dependency scores were calculated for each cell line as in Figure 2. Note that most cancer cells with the highest ARID1B dependency (lower than 0.5) have ARID1A mutations.

**Table 1 cells-12-00952-t001:** Frequency of ARID1A deficiency in various malignancies.

Types of Cancer	Mutation Rate	Reference
Ovarian clear cell carcinoma	40–57%	[15]
High-grade endometrioid adenocarcinoma	60%	[16]
Low-grade endometrioid adenocarcinoma	47%	[16]
Ovarian endometrioid carcinomas	30%	[15]
Cholangiocarcinomas, intrahepatic	7–36%	[17]
Cholangiocarcinomas, extrahepatic	5–12%	[17]
Gastric adenocarcinomas	18–27%	[18]
Hepatocellular carcinoma	10–17%	[19]
Pancreatic ductal adenocarcinoma	6–10%	[20]
Colon cancer	10%	[21]
Lung cancer	8%	[22]
Neuroblastoma	6%	[23]

**Table 2 cells-12-00952-t002:** Recent clinical trials of ARID1A-deficient cancers.

Title	Cancer/Stage	Drug/Combination	Phases	Status	NCT#
Phase II Study of Tazemetostat in Solid Tumors Harboring an ARID1A Mutation	Solid Tumor|ARID1A Gene Mutation	Drug: Tazemetostat	Phase 2	Recruiting	NCT05023655
PD-1 Combined with Dasatinib for as Third-line Treatment for ARID1A Mutation Advanced NSCLC	NSCLC Stage IV|ARID1A|PD-1	Drug: PD-1 plus Dasatinib	Phase 2	Unknown status	NCT04284202
Bevacizumab and/or Niraparib in Patients with Recurrent Endometrial and/or Ovarian Cancer With ARID1A Mutation	Recurrent Endometrial Carcinoma|Recurrent Ovarian Carcinoma|ARID1A Gene Mutation	Drug: Bevacizumab|Drug: Niraparib	Phase 2	Not yet recruiting	NCT05523440
ATr Inhibitor in Combination with Olaparib in Gynaecological Cancers With ARId1A Loss or no Loss	Gynaecological Cancers	Drug: AZD6738|Drug: Olaparib	Phase 2	Recruiting	NCT04065269
Nivolumab for the Treatment of Metastatic or Unresectable Solid Tumors with ARID1A Mutation and CXCL13 Expression	Metastatic Malignant Solid Neoplasm|Unresectable Solid Neoplasm	Biological: Nivolumab	Phase 2	Recruiting	NCT04957615
Nivolumab for the Treatment of Patients with Metastatic Urothelial Cancer With ARID1A Mutation and Stratify Response Based on CXCL13 Expression	Multiple cancers	Other: Diagnostic Laboratory Biomarker Analysis|Biological: Nivolumab	Phase 2	Not yet recruiting	NCT04953104
Evaluating Safety & Efficacy Belinostat Combo w Nivo Alone & w Ipi in Patients w Treated Metastatic/Advanced Carcinomas w ARID1A Lof Mutation	Metastatic Adenocarcinoma	Drug: Belinostat|Drug: nivolumab|Drug: ipilimumab	Phase 1	Withdrawn	NCT04315155
Tremelimumab, Durvalumab, and Belinostat for the Treatment of ARID1A Mutated Metastatic or Unresectable, Locally Advanced Urothelial Carcinoma	Multiple cancers	Drug: Belinostat|Biological: Durvalumab|Biological: Tremelimumab	Phase 1	Recruiting	NCT05154994
PLX2853 as a Single Agent in Advanced Gynecological Malignancies and in Combination with Carboplatin in Platinum-Resistant Epithelial Ovarian Cancer	Gynecologic Neoplasms|Epithelial Ovarian Cancer	Drug: PLX2853|Drug: Carboplatin	Phase 1|Phase 2	Terminated	NCT04493619
Dasatinib in Treating Patients with Recurrent or Persistent Ovarian, Fallopian Tube, Endometrial or Peritoneal Cancer	Multiple cancers	Drug: Dasatinib|Other: Laboratory Biomarker Analysis	Phase 2	Active, not recruiting	NCT02059265
JAB-2485 Activity in Adult Patients with Advanced Solid Tumors	Multiple cancers	Drug: JAB-2485 (Aurora A inhibitor)	Phase 1|Phase 2	Not yet recruiting	NCT05490472
Olaparib in Treating Patients with Metastatic Biliary Tract Cancer With Aberrant DNA Repair Gene Mutations	Multiple cancers	Drug: Olaparib	Phase 2	Recruiting	NCT04042831
Prognostic Biomarkers in Patients with Urothelial Carcinoma	Bladder Cancer			Completed	NCT04872036
A Study of PLX2853 in Advanced Malignancies.	Multiple cancers	Drug: PLX2853	Phase 1	Completed	NCT03297424

## Data Availability

Not applicable.

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
