# Peer review of "Targeting ARID1A-Deficient Cancers: An Immune-Metabolic Perspective"

_cells, 2023, doi:10.3390/cells12060952_

Round 1
Reviewer 1 Report
Suggestion: if authors are willing to revise it by following the PRISMA guidelines and do a proper systematic review, this can be considered. In particular, they need to address the key areas such as number of studies included for analysis, key results or findings from this review, and study limitations, etc. Check the PRISMA guidelines for systematic review and meta-analysis..
-New references for 2021 and 2022 will be added to the introduction and Discussion
-Abbreviations should have been explained at least once.
-The first time you include acronyms within the text, you have to write them in full. After that, you should report them as abbreviations only.
-Complete all the references in accordance with the Instructions for the authors
-Separate the result and the discussion
-Please include the strengths and weaknesses of each study that is presented
Author Response
Reviewer #1
General comment:
Suggestion: if authors are willing to revise it by following the PRISMA guidelines and do a proper systematic
review, this can be considered. In particular, they need to address the key areas such as number of studies
included for analysis, key results or findings from this review, and study limitations, etc. Check the PRISMA
guidelines for systematic review and meta-analysis.
We are grateful for the reviewer’s comment and we appreciate it. The PRISMA guidelines have been designed
primarily for systematic reviews of studies that evaluate the benefits and harms of a health care intervention by
providing a transparent, complete, and accurate account of why the review was done, what they did, and what
they found. Evidence from observational studies indicates that the use of these guidelines is associated with more
complete reporting of systemic reviews (Page MJ, McKenzie JE, Bossuyt PM, Boutron I, Hoffmann TC, Mulrow
CD, Shamseer L, Tetzlaff JM, Akl EA, Brennan SE, Chou R, Glanville J, Grimshaw JM, Hróbjartsson A, Lalu
MM, Li T, Loder EW, Mayo-Wilson E, McDonald S, McGuinness LA, Stewart LA, Thomas J, Tricco AC,
Welch VA, Whiting P, Moher D. The PRISMA 2020 statement: an updated guideline for reporting systematic
reviews. BMJ. 2021,372,n71). Given that the current review manuscript is focused on the immune-metabolic
mechanisms of ARID1A-deficient cancers, a better understanding of this emerging molecular vulnerability
elicited by ARID1A mutations may open up new avenues for more effective clinical interventions to improve
patient outcome. Notably, there are currently a number of ongoing clinical trials in ARID1A-deficient cancers
(Table 2 in the revised manuscript). But a systemic literature of clinical observations of these interventions are
not available yet. Once these data are released, a comprehensive review and meta-analysis would be helpful to
evaluate the clinical benefits and harms of these therapeutic strategies by following the PRISMA guidelines.
Specific comments:
(1) New references for 2021 and 2022 will be added to the introduction and Discussion
We thank the reviewer’s suggestions. We have updated the Introduction and Discussion and have included two
2023 articles in the Introduction (Ref. 5 in the revised manuscript) and Discussion (Ref. 78 in the revised
manuscript), respectively.
(2) Abbreviations should have been explained at least once.
We apologize for the omission. We have provided the full name of abbreviations in the revised manuscript the
first time when they appear in the text.
(3) The first time you include acronyms within the text, you have to write them in full. After that, you should
report them as abbreviations only.
We apologize for the confusion. We have provided the full name of acronyms in the revised manuscript the first
time when they appear, and then we have used abbreviations thereafter throughout the text.
(4) Complete all the references in accordance with the Instructions for the authors
We are grateful for the reviewer’s comment. We have revised the format of the references according to the
Instructions for the Authors.
(5) Separate the result and the discussion
We thank the reviewer’s comment. We have revised the text and have made the separation between the result and
discussion as clear as possible.
(6) Please include the strengths and weaknesses of each study that is presented
We thank the reviewer’s comment. In our manuscript, we have critically analyzed the studies that are included in
our review. In particular, we have highlighted the seemingly controversial results of certain studies, for example,
studies from Ref. 52, 53 and 54 and those from Ref. 56 and 57, which we believe is in line with the reviewer’s
comment.
Reviewer 2 Report
This is a nice review to provide the role of ARID1A deficiency in cancer in terms of immuno-metabolic perspectives. However, several issues needed to be addressed before further consideration for publication.
Major:
1. The frequency of ARID1A deficiency among various cancer types should be provided as a figure or table to give a background facilitating readers to follow the further immuno-metabolic perspectives.
2. ARID1A deficiency influences energy metabolism in lung cancer should be mentioned. the reference is "Chromatin Remodeling Induced by ARID1A Loss in Lung Cancer Promotes Glycolysis and Confers JQ1 Vulnerability". "Mutations in the SWI/SNF complex induce a targetable dependence on oxidative phosphorylation in lung cancer".
3. In the immunity modulator part, several articles should be discussed as well. It will improve the comprehensiveness of the immune context regarding ARID1A deficiency.
a.The clinical significance of PD-L1 in advanced gastric cancer is dependent on ARID1A mutations and ATM expression.
b.ARID1A loss induces polymorphonuclear myeloid-derived suppressor cell chemotaxis and promotes prostate cancer progression.
Minor:
1. page 3, line 93,98 please provide the mutation rate of ARID1A as one of the most frequently mutated gene.
2.Figure 2, please define and explain how to calculate the ARID1A dependency score. Please also interpret what is speculated for the association between ARID1A mutation and less ARID1A depletion.
3. Similarly, in Figure 3, please provide the materials and methods to define the ARID1B dependency score. Is the score more negative indicating more dependent on ARID1B?
4. The outline showed the absence of section 5 before the final section "future outlooks". Please check if any text is missing.
5. Page 5, line 204 "ARID1D" please check the spelling.
Author Response
Reviewer #2
General comment:
This is a nice review to provide the role of ARID1A deficiency in cancer in terms of immuno-metabolic
perspectives. However, several issues needed to be addressed before further consideration for publication.
We appreciate the reviewer’s comment. We have provided more information to address the issues raised by the
reviewer.
Specific comments:
(1) The frequency of ARID1A deficiency among various cancer types should be provided as a figure or table
to give a background facilitating readers to follow the further immuno-metabolic perspectives.
We are grateful to the reviewer’s comment. We have added Table 1 in the revised manuscript to provide the
mutation frequency of ARID1A deficiency in various cancers.
(2) ARID1A deficiency influences energy metabolism in lung cancer should be mentioned. the reference is
"Chromatin Remodeling Induced by ARID1A Loss in Lung Cancer Promotes Glycolysis and Confers JQ1
Vulnerability". "Mutations in the SWI/SNF complex induce a targetable dependence on oxidative
phosphorylation in lung cancer".
We apologize for omitting these two articles and we are grateful to the reviewer’s comment. We have included
the key findings from "Chromatin Remodeling Induced by ARID1A Loss in Lung Cancer Promotes Glycolysis
and Confers JQ1 Vulnerability" (Ref. 61 in the revised manuscript) and "Mutations in the SWI/SNF complex
induce a targetable dependence on oxidative phosphorylation in lung cancer" (Ref. 19 in the revised manuscript)
in Section 4.1 “Metabolic rewiring in ARID1A-deficient cancers”.
(3) In the immunity modulator part, several articles should be discussed as well. It will improve the
comprehensiveness of the immune context regarding ARID1A deficiency.
a.The clinical significance of PD-L1 in advanced gastric cancer is dependent on ARID1A mutations and
ATM expression.
b.ARID1A loss induces polymorphonuclear myeloid-derived suppressor cell chemotaxis and promotes
prostate cancer progression.
We apologize for the omission and we are grateful to the reviewer’s comment. We have included the major
findings from "a. The clinical significance of PD-L1 in advanced gastric cancer is dependent on ARID1A
mutations and ATM expression" (Ref. 65 in the revised manuscript) and " b. ARID1A loss induces
polymorphonuclear myeloid-derived suppressor cell chemotaxis and promotes prostate cancer progression"
(Ref. 77 in the revised manuscript) in Section 4.2 “ARID1A as a modulator of antitumor immunity”.
(4) page 3, line 93,98 please provide the mutation rate of ARID1A as one of the most frequently mutated
gene.
Please see the response to Specific comment #1.
(5) Figure 2, please define and explain how to calculate the ARID1A dependency score. Please also interpret
what is speculated for the association between ARID1A mutation and less ARID1A depletion.
We apologize for the confusion. In the revised Figure 2 legend, we have included a detailed description of how
the dependency score is calculated, as well as the interpretation of the dependency score to explain the
association between the mutation status of ARID1A and the dependency on ARID1A depletion.
(6) Similarly, in Figure 3, please provide the materials and methods to define the ARID1B dependency score.
Is the score more negative indicating more dependent on ARID1B?
We are grateful for the reviewer’s comment. In the revised manuscript, we have added a detailed description of
the materials and methods used to define the ARID1B dependency score in Section 3 “ARID1A: synthetic
lethality and beyond”. We have also provided a detailed interpretation of the dependency score. Furthermore, we
have included a Supplemental Table 1 to show the dependency on ARID1B in a wide variety of human cancers
carrying ARID1A mutations.
(7) The outline showed the absence of section 5 before the final section "future outlooks". Please check if any
text is missing.
We thank the reviewer for pointing it out and we have fixed the error in the revised manuscript.
(8) Page 5, line 204 "ARID1D" please check the spelling.
We thank the reviewer for pointing it out and this has been fixed in the revised manuscript.
Round 2
Reviewer 2 Report
Authors have fixed all the concerns. Still need a spelling check and minor revision for grammar editing.
Author Response
Reviewer #2
Authors have fixed all the concerns. Still need a spelling check and minor revision for grammar editing
We are grateful for the reviewer’s comment and we appreciate it. In the revised manuscript, we have carefully checked spelling and grammar throughout the manuscript.